# Surgery for T1/2N0 Oropharyngeal Carcinoma Is a Better Treatment Option than Radiotherapy—A Long-Term Follow-Up Study from a Single Japanese High-Volume Cancer Center

**DOI:** 10.3390/cancers17111862

**Published:** 2025-05-31

**Authors:** Masahiro Rokugo, Takeshi Shinozaki, Ryo Ishii, Yusuke Ito, Shingo Sakashita, Genichiro Ishii, Takenori Ogawa, Yukio Katori, Kazuto Matsuura, Ryuichi Hayashi

**Affiliations:** 1Department of Head and Neck Surgery, National Cancer Center Hospital East, Kashiwa 277-8577, Chiba, Japan; masahiro.rokugo.c1@tohoku.ac.jp (M.R.); tashinoz@east.ncc.go.jp (T.S.); yusuke19881988@gmail.com (Y.I.); kmatsuur@east.ncc.go.jp (K.M.); 2Department of Otolaryngology-Head and Neck Surgery, Tohoku University Graduate School of Medicine, Sendai 980-8574, Miyagi, Japan; rishii@orl.med.tohoku.ac.jp (R.I.); yukio.katori.d1@tohoku.ac.jp (Y.K.); 3Division of Pathology, Exploratory Oncology Research and Clinical Trial Center, National Cancer Center East, Kashiwa 277-8577, Chiba, Japan; ssakashi@east.ncc.go.jp; 4Department of Pathology and Clinical Laboratories, National Cancer Center Hospital East, Kashiwa 277-8577, Chiba, Japan; gishii@east.ncc.go.jp; 5Department of Otolaryngology-Head and Neck Surgery, Gifu University Graduate School of Medicine, Gifu 501-1194, Gifu, Japan

**Keywords:** oropharyngeal cancer, squamous cell carcinoma, surgery

## Abstract

This retrospective study included 94 cases of early-stage (T1/2N0) oropharyngeal squamous cell carcinoma treated with surgery or radiotherapy. Of these, 74 cases underwent surgery, and 20 cases received radiotherapy. Of the patients who underwent surgery, 57 underwent transoral surgery, and 17 underwent pharyngectomy via transcervical approach. The five-year overall survival was significantly higher in the surgery group compared to radiotherapy (74.9% vs. 51.0%, *p* = 0.035). Disease-specific survival and local control rates also favored surgery but did not reach statistical significance. Notably, 57.4% of cases suffered from multiple primary cancers, of which 30.9% were in the head and neck region and all were metachronous. These findings suggest that surgery is an effective initial treatment for early-stage oropharyngeal cancer, which often involves metachronous multiple primary cancers, and contributes to improving prognosis.

## 1. Introduction

Head and neck cancer was the eighth most common cancer worldwide in 2020 (745,000 new cases and 365,000 deaths) [1]. Although the incidence of oropharyngeal squamous cell carcinomas (OPSCCs) has increased substantially, especially in developed countries, the overall annual incidence of squamous cell carcinoma of the head and neck (HNSCC) has decreased substantially [2,3,4,5]. The American Joint Committee on Cancer/Union for International Cancer Control (AJCC/UICC) released the *TNM Classification of Malignancies, 8th Edition*, on 1 January 2018, dividing oropharyngeal cancer into p16-positive and p16-negative tumors [6]. The expression of the tumor suppressor protein cyclin-dependent kinase inhibitor 2A (p16-INK4a, p16) serves as a surrogate marker for human papillomavirus-induced tumors [7]. The incidence of HPV-associated OPSCCs is increasing [8,9], and many studies have shown that compared with smoking-related and alcohol-related OPSCCs, HPV positivity is associated with an improved prognosis, which is why the UICC 8th edition has a new classification for p16-positive OPSCCs [10,11,12,13].

According to the National Comprehensive Cancer Network guideline (NCCN guideline, ver.2.2025), T1/2N0 OPSCCs should ideally be treated with a single modality therapy, either primary surgery with or without elective neck dissection (ND) or definitive radiotherapy (RT), regardless of p16 status. In recent years, transoral robotic surgery (TORS) has emerged as a viable and minimally invasive surgical option for OPSCCs, demonstrating favorable oncological and functional outcomes, particularly in HPV-associated tumors [14]. However, the complex functions of the oropharynx, such as swallowing and articulation, and anatomical access difficulties, such as the root of the tongue, are potential problems in treating OPSCCs. Therefore, despite these advances, no conclusion has been reached regarding the optimal treatment option for T1/2N0 OPSCCs. RT remains popular for early OPSCCs because it provides a nonsurgical opportunity with equivalent oncological outcomes [15,16,17,18,19,20,21]. However, RT can also seriously impact long-term functional outcomes, including pharyngeal fibrosis, xerostomia, loss of taste, osteoradionecrosis, trismus, aspiration, and quality of life [22,23,24]. Surgery alone makes it possible to prevent the acute and late-onset adverse events of RT. Additionally, recent studies highlight the advantages of TORS in reducing morbidity compared to conventional surgery and RT, emphasizing improved swallowing function and quality of life [25]. Because head and neck cancers, including OPSCCs, often have simultaneous and/or metachronous multiple primary cancers at high frequency, an optimal long-term therapeutic strategy should be identified [26]. The proportion of multiple cancers occurring in the head and neck region does not change when comparing HPV-positive or HPV-negative oropharyngeal cancers [10]. In addition, surgery may preserve radiation therapy options for another head and neck cancer that may occur in the future.

Our department often follows up with patients for longer than five years after initial treatment. P16 status has been mentioned since the 8th edition of the TNM classification was released in 2018, but it has only been a few years since the category was introduced, and there are few reports of long-term follow-up with a focus on early-stage OPSCCs. Moreover, the adoption of robotic-assisted techniques such as TORS has increased dramatically during this period, altering the treatment landscape for OPSCCs. However, there remains limited evidence directly comparing the long-term outcomes between traditional surgical techniques, TORS, and RT specifically for early-stage OPSCCs. Therefore, the purpose of this study was to compare the long-term prognosis and incidence of multiple cancers in early-stage OPSCCs after initial treatment by p16 status and treatment method. We hypothesized that surgery as the initial therapy for T1/2N0 OPSCCs results in higher survival rates than radiation therapy, preserving treatment options for future multiple cancers. Given recent developments, we also aimed to assess our results in the context of emerging evidence supporting the use of TORS. This study aims to determine the frequency of multiple cancers after treating OPSCCs and examine appropriate initial treatment for T1/2N0 OPSCCs.

## 2. Materials and Methods

### 2.1. Patients

We conducted a single-center retrospective study at Japan National Cancer Center Hospital East (Chiba, Japan), a high-volume center in Japan. The inclusion criteria were as follows: (1) patients who were diagnosed with histologically confirmed OPSCCs between January 2000 and December 2012; (2) patients who were diagnosed with primary lesions as T1 or T2; (3) patients who exhibited no cervical node metastasis detected by physical and radiological findings; and (4) patients who underwent surgery or RT as the initial treatment for OPSCCs.

Patients who were incapable of general anesthesia, patients indicated for palliative RT, and patients with ECOG Performance Status (PS) Scale scores of 2 or higher were excluded.

### 2.2. Evaluation

We discussed all the cases before planning treatment policy at a multidisciplinary tumor board. For clinical staging, we used the TNM classification of the version at that time (5th edition for cases in 2000–2001, 6th for 2002–2008, 7th for 2009–2012). Still, these cases were newly diagnosed according to the 8th edition for this study. The significant difference between the 8th and earlier editions is that T2N0 in p16-positive OPSCCs is downgraded from stage II to stage I, while the T1 and T2 definitions and settings remain unchanged.

The oropharynx was classified into four subsites according to the *General Rules for Clinical and Pathological Studies on Head and Neck Cancer, 6th edition (revised edition)*, which is consistent with international standards such as the AJCC and NCCN guidelines. The anterior wall includes the base of the tongue and the vallecula. The lateral wall includes the palatine tonsil, tonsillar fossa, and palatoglossal arch. The posterior wall refers to the posterior pharyngeal wall. The superior wall consists of the inferior surface of the soft palate and the uvula.

p16 status was used as a surrogate marker for HPV positivity, and tumors were classified as p16-positive in the case of solid and diffuse nuclear and cytoplasmic staining in >70% of tumor cells, as evaluated by immunohistochemistry. In this study, p16 immunohistochemical staining was performed on eligible patients’ surgical or biopsy pathology sections. Positive and negative p16 determinations were made by our pathologist, who specializes in the pathology of head and neck tumors. In addition, in cases where immunohistochemical staining could not be performed, the stage was diagnosed according to the pre-7th. Among the 94 cases, p16 immunohistochemical staining was performed in 83 patients. In 11 patients, p16 status could not be determined due to unavailable or inadequate pathological samples.

In this study, surgical margins were categorized into horizontal and vertical components. Horizontal margins were defined as mucosal resection surfaces, including anterior, posterior, superior, and lateral edges. Vertical margins were defined as the deep resection margins, and their anatomical direction depended on the tumor subsite:Anterior wall tumors: intrinsic/extrinsic tongue musculatureLateral wall tumors: pharyngeal constrictor muscles or prevertebral fasciaSuperior wall tumors: soft palate musculature or palatal aponeurosisPosterior wall tumors: posterior pharyngeal wall musculature or prevertebral fascia

Multiple primary cancers that occurred after five years or more were considered metachronous, even if they occurred at the original site.

### 2.3. Treatment

Patients were selected for either primary surgery or primary RT. The treatment strategy was as follows: (1) transoral surgery (TOS) was prioritized over RT or pharyngectomy via transcervical approach (open resection); and (2) TOS was performed without using surgical robots or endoscopy. Patients chose either open resection or RT as the course of treatment. The surgical approach was selected according to the location of the primary tumor (partial oropharyngectomy, tonsillectomy). Selective neck dissection was performed simultaneously in some pharyngotomies via open resection and some transoral surgeries. Free-flap reconstruction was also performed after open resection, if necessary.

In cases where recurrence was strongly suspected, clinically based on intraoperative diagnosis and postoperative pathology results, adjuvant RT with concurrent chemotherapy (POCRT) was performed according to NCCN guidelines.

The patient underwent X-ray therapy or proton beam therapy. RT was administered as intensity-modulated radiation therapy (IMRT) using simultaneous integrated boost (SIB) or three-dimensional radiation therapy (3D-RT) or two-dimensional radiation therapy (2D-RT).

Surgical method and RT cases were not evenly distributed in some periods.

### 2.4. Follow-Up

Patients were evaluated every 1–2 months up to two years after completion of initial treatment, every three months in the third year, every six months in the fourth and fifth years, and every year after the sixth year. We routinely performed physical examinations, flexible endoscopy, and CT or MRI when imaging was needed. Follow-up imaging (CT or MRI) was completed three months after the initial treatment, and every three months in the first year after the end of treatment. And again, imaging was performed every 3–6 months for the second year and every six months to one year after the third year. Biopsy was also performed if recurrence was clinically suspected or if there were multiple cancerous lesions. If multiple cancers were diagnosed, we began our follow-up on the same schedule starting at that time.

### 2.5. Statistical Analysis

All continuous variables were compared using Student’s *t*-test, and categorical variables were compared using the chi-squared or Fisher’s exact test if events were minor. We compared overall survival (OS), disease-specific survival (DSS), and local control rates (LC) using the Kaplan–Meier method and log-rank test (Mantel–Cox). Statistical tests were 2-tailed with an α level of 0.05. Multivariate analyses were performed using the Cox proportional hazards model. One patient with a follow-up duration of only 48 days was included in the survival analysis. This patient was alive at the last follow-up and was appropriately right-censored. As per the standard Kaplan–Meier methodology, right-censored data do not affect the estimation of survival probabilities. All statistical analyses were performed using JMP Pro 17 (SAS Institute Inc., Cary, NC, USA). Figures were generated using GraphPad Prism version 10.4.1 (GraphPad Software, San Diego, CA, USA), and tables were created using Microsoft Excel for Microsoft 365 (Microsoft Corporation, Redmond, WA, USA).

## 3. Results

We identified 94 patients who met these criteria. The median follow-up period was 2131 days (approximately 5.8 years), with a range of 49 to 6945 days (approximately 0.1 to 19.0 years). The mean follow-up period was 2378 ± 1694 days (approximately 6.5 ± 4.6 years). The median age was 68 years overall. The mean age was 68 ± 9.6 years. When stratified by treatment group, the surgery group had a median age of 66.5 years (range: 35–82), while the RT group had a higher median age of 77.5 years (range: 49–92) (Table 1). There were 80 males and 14 females. Thirty-four patients had T1N0 disease, and 60 had T2N0 disease. Surgery was performed on 74 patients. RT was performed on 20 patients, 16 of whom (80%) had T2 disease, indicating a higher proportion of T2 in the RT group.

Among the 94 patients, p16 immunohistochemical staining was successfully performed in 83 cases, of which 22 (26.5%) were positive and 61 (73.5%) were negative. In the remaining 11 cases (11.7%), staining could not be performed due to insufficient or unavailable specimens. Sixteen of the 22 (72.7%) p16-positive cases were T2. According to the AJCC/UICC TNM 8th, 50 cases were Stage I and 44 were Stage II. With the AJCC/UICC TNM classification switch from pre-7th to 8th, these 16 cases of p16-positive T2N0 were downgraded from stage II to stage I.

The subsites were the lateral wall in 53 patients, the anterior wall in 12, the superior wall in 22, and the posterior wall in 7.

Details of the 74 patients who underwent surgery are summarized in Table 2: 54 patients (73.0%) underwent transoral surgery (TOS) only, 3 patients (4.0%) underwent TOS with ND, and 17 patients (23.0%) underwent open resection. Sixteen of the 17 patients who had an open resection underwent concurrent neck dissection (ND); in the remaining one case, the tumor extended from the base of the tongue to the vallecula. A transcervical approach was used for resection, and partial removal of the left side of the epiglottis was performed to ensure adequate surgical margins. Nine of the 17 patients (52.9%) who had an open resection underwent free-flap reconstruction. Regarding postoperative adjuvant therapy, only one patient who underwent an open resection with free-flap reconstruction for a T2 tumor of the upper wall received POCRT.

Of the 74 patients who underwent surgery, positive surgical margins were detected in 32 cases (43.2%) (horizontal in 27 cases, vertical in 2 cases, and 3 cases with both). Nine of these 32 cases (28.1%) demonstrated local recurrence (7 out of 27 cases (25.9%) in the horizontal margin, 1 out of 2 (50.0%) in the vertical margin, and 1 out of 3 (33.3%) in both margins. The median time to local recurrence among these 15 patients was 1375 days (range: 168–4081), and the mean was 1434 days.

The total median radiation dose for the 20 patients who received RT was 66 Gy (66–80 Gy). One patient underwent proton beam therapy, while 19 patients underwent X-ray therapy. Fourteen patients underwent 3D irradiation, one patient underwent 2D irradiation, and only 5 underwent intensity-modulated radiation therapy (IMRT). No chemotherapy was administered to any of the patients in the RT group.

All patients’ 5-year overall survival (OS) rate and 10-year OS were 70.2% and 61.2%. The 5-year OS was 74.9% and 51.0% for the surgery and RT groups, respectively (Figure 1a), and the 10-year OS was 67.0% and 34.0% for the surgery and RT groups, respectively. The surgery group had a significantly higher OS (*p* = 0.036).

All patients’ 5-year disease-specific survival (DSS) rate and 10-year DSS were 80.5% and 77.4%. The 5-year DSS was 86.0% and 64.5% for the surgery and RT groups, respectively (Figure 1b), and the 10-year DSS was 80.5% and 64.5% for the surgery and RT groups, respectively. The surgery group seemed to provide a higher DSS, but there were no statistically significant differences (*p* = 0.116).

The 5-year local control (LC) rates were 76.5% and 59.1% for the surgery and RT groups, respectively (**c**). The 10-year LC rates were 72.6% and 59.1% for the surgery and RT groups. The surgery group appeared to have a higher LC, although it did not indicate a statistically significant difference (*p* = 0.106). More than half of the cases of recurrence involved local recurrence within 500 days. LC rates did not differ by clinical stage or subsite. No disease-specific primary death or local recurrence occurred after the 10th year.

We then created Kaplan–Meier plots for the 83 cases, excluding the 11 cases in which p16 staining was impossible, and divided them into four groups according to p16 status and treatment. Although the DSS did not show significant differences, the OS and LC results showed significantly different behavior in the p16-negative RT treatment group compared to the other three groups (Figure 2a–c). The inferiority of RT over surgery in OS (Figure 1a) may be due to the fact that RT was performed on p16-negative OPSCCs.

Table 3 shows the relationship between the initial treatment outcome and the salvage treatments after the first recurrence. Ten cases after TOS recurred in the primary lesion (rT), and eight of those cases (80%) underwent salvage surgery (seven cases) or definitive RT (one case). In contrast, seven cases after open resection recurred in the primary lesion, and five of those cases (83.3%) underwent salvage surgery. Finally, seven cases after initial RT recurred in the primary lesion, and six of those cases (85.7%) underwent salvage surgery. Regarding regional lymph node recurrence (rN), a total of 14 cases after surgery recurred, and all of them underwent salvage surgery (13 cases) or definitive RT (one case). No distant metastasis (rM) cases were found in the initial recurrence.

Actual outcomes following the initial treatment were also investigated (Table 4). The 64 surviving patients comprised 68.1% of the whole cohort. There were no cancer-bearing survivors. Of the patients who died of their original disease, the most common cause was due to the primary lesion (*n* = 11). Of the seven cases where the patient underwent surgery for primary recurrence after RT, five died (83.3%) with the original cancer. In contrast, four of the thirteen cases (30.8%) who underwent salvage surgery or definitive RT for primary recurrence after surgery died due to the primary lesion (Table 3). There was also a high proportion of patients who died of other cancers or diseases. Two died due to other malignancies, including a metachronous head and neck malignancy in one patient. The other two cases died of different conditions, including pneumonia in one patient. There were no treatment-related deaths. On the final follow-up day of this study, all patients could take food orally, and no patients required long-term tracheotomy retention.

The results of the univariate analysis for overall survival are shown in Table 5, and the results of the multivariate analysis are in Table 6. These analyses were performed on 83 cases, excluding 11 cases in which p16 staining was difficult concerning the p16 status and stage factors. In univariate analysis, the treatment method and the presence of multiple cancers were statistically significant prognostic factors in the overall survival rate (*p* = 0.041 and *p* = 0.025, respectively). As many previous reports have shown, p16 status impacts the prognosis of oropharyngeal cancer patients [10,11,12,13]. Multivariate analysis showed that the treatment method was the only significant prognostic factor (HR 2.75, 95% CI 1.23–6.16, *p* = 0.014) (Table 6). None of the other two parameters showed a significant difference. Next, to assess the potential confounding effect of age and the presence of multiple primary cancers, we built an additional multivariable Cox model that included age and multiple-cancer status together with treatment modality, p16 status, and 8th-edition stage. In this expanded model, age (HR = 1.03, 95% CI 0.98–1.07, *p* = 0.283) and multiple cancers (HR = 1.97, 95% CI 0.82–4.76, *p* = 0.131) were not significant independent predictors of OS, whereas the detrimental effect of RT versus surgery persisted (HR: 2.62, 95% CI 1.13–6.06, *p* = 0.024), suggesting a persistent trend toward worse survival with RT. These findings support the robustness of our primary conclusion while acknowledging the influence of additional covariates.

Multiple primary cancers were found in 54 of the 94 patients (57.4%). Of those 54 patients, 13 (24.1%) had multiple primary cancers in the head and neck region only, 16 (29.6%) had multiple primary cancers in the head and neck region and other regions, and 25 had multiple primary cancers in other regions only. Of the 29 patients (30.9% of all cases) with multiple primary cancers in the head and neck region, 24 were p16-negative (24 out of 29: 82.8%), three were p16-positive, and two had unknown p16 staining results. All of the 29 patients who suffered from multiple malignancies in the head and neck region had only metachronous multiple primary cancers. Table 7 shows details of the sites where the second cancers were found. In the head and neck region, hypopharyngeal cancer was the most common, followed by oral cancer, nasopharyngeal cancer, laryngeal cancer, and sinonasal cancer. The most common site for cancer outside the head and neck region was the esophagus, followed by the stomach, lungs, and colon.

Since a single patient may have more than one primary cancer in addition to oropharyngeal cancer, the numbers represent the total number of cancers.

## 4. Discussion

Due to the nature of head and neck cancers, in which patients are prone to multiple cancers in the head and neck region, we have shown that surgery may be a better survival option than RT, when considering long-term strategy, as the initial treatment for T1/2N0 OPSCCs. Previous reports have reported a 5-year OS of 73–79% when surgery is selected as the initial treatment for T1/2N0 OPSCCs, while 5-year OS was 43–83%/39–68% for T1/2 OPSCCs when RT was chosen as the initial treatment [17,27,28,29,30,31,32]. Our OS results (74.9% and 51.0% for the surgery and RT groups, respectively) are comparable to the previous literature examining treatment for T1/T2N0 OPSCCs. This is the first report of long-term survival of T1/2N0 OPSCCs from a Japanese high-volume center for more than ten years. Most previous reports have found no significant difference in outcomes between surgery and RT at 3 or 5 years of OS [15,16,17,18,19,20,21]. However, in their review article, Michael et al. concluded that initial surgical treatment yielded better survival rates than RT treatment in T1/T2N0 lateral wall OPSCCs [32]. Similarly, this study shows that the surgical group’s OS was higher than the RT group’s. This result may be influenced by the survival outcome of p16-negative oropharyngeal carcinoma (Figure 2a–c). In addition to p16 status, the age difference between the groups may have contributed to the observed OS discrepancy. Patients in the RT group were significantly older than those in the surgery group (median 77.5 vs. 66.5 years), which likely reflects a clinical tendency to select RT for older or potentially more comorbid patients. However, it should be noted that patients with an ECOG Performance Status score of 2 or higher were excluded from the study. Therefore, all included patients were deemed fit for curative-intent treatment. Furthermore, a supplementary multivariable Cox regression model included age and treatment modality, p16 status, stage, and occurrence of multiple primary cancers. In this expanded model, age was not a statistically significant predictor of OS (*p* = 0.283), suggesting that age alone may not have been a major confounding factor in the observed survival differences. While comorbidity data were not uniformly available and remain a limitation, these findings support the conclusion that the observed OS benefit in the surgery group is unlikely to be solely explained by baseline age differences.

Although residual confounding due to unmeasured comorbidities cannot be entirely excluded, the consistent trend in favor of surgery, despite adjustment for age, stage, p16 status, and multiple cancers, highlights its potential oncologic benefit. The observed OS difference may partially reflect non-cancer-related mortality in the RT group, or unaccounted treatment complexity such as hyperfractionation or chemoradiotherapy. The small sample size of the RT group may also contribute to selection bias in this retrospective study. Nonetheless, these findings support the interpretation that surgery provides a robust oncologic advantage in early-stage OPSCCs, particularly when long-term outcomes and treatment-related sequelae are considered. Over the past decade, TORS has become increasingly adopted as a minimally invasive surgical approach, with multiple studies demonstrating its efficacy in achieving high locoregional control rates and excellent functional preservation, especially in HPV-positive tumors [14,25]. Notably, recent analyses have shown that TORS is associated with reduced rates of gastrostomy dependence, improved swallowing function, and shorter hospital stays compared to conventional open surgery and RT [33]. These data support the relevance of surgical approaches, particularly TORS, in treating T1/2N0 OPSCCs, aligning with our findings. Given the functional advantages and the opportunity to avoid radiation-induced toxicities, TORS may represent an optimal first-line modality in appropriately selected patients.

This study suggests that the primary treatment option for T1/2N0 OPSCCs should be surgery, with RT being reserved as a later therapeutic option. In particular, surgery is proactively considered a choice when it is anticipated that the cancer can be resected entirely orally, or in cases of cancer in the lateral wall with tongue base infiltration, mimicking T4a due to invasion of the deep muscle of the tongue, which is considered to have poor radiation sensitivity. With the growing use of TORS, these anatomically challenging cases may now be more accessible surgically, further reinforcing the utility of primary surgery even in borderline resectable tumors.

When discussing treatment options, the main focus should be on the functional outcomes, prevention of late complications, and the risk of second primary tumor development in a previously irradiated area. While RT techniques have tremendously improved over the past decades and have significantly reduced late toxicity effects, these issues are still relevant and deserve special consideration. Long-term complications of RT for the head and neck fields, such as pharyngeal fibrosis, xerostomia, osteoradionecrosis, trismus, aspiration, or ischemic stroke, are reported [22,23,24,34]. Some degree of trismus is seen in almost every patient treated for OPSCCs, regardless of the treatment modality [35,36]. However, the incidence is higher in patients receiving RT than in surgery [34]. Importantly, TORS has been shown to significantly reduce the incidence of long-term dysphagia and dependency on feeding tubes, while preserving speech and swallowing, which are crucial quality-of-life outcomes for patients with early-stage OPSCCs [25]. These considerations become particularly critical in younger patients with a long life expectancy. The development of multiple primary malignancies in head and neck cancer patients, particularly upper aerodigestive tract carcinomas, is relatively high [37,38]. In this investigation, multiple primary cancers occurred in 57.4% of T1/2N0 OPSCCs. In particular, the incidence of multiple primary cancers in the head and neck region was 30.9%. This study was limited to early-stage OPSCCs with good prognosis, which may have resulted in more prolonged survival and a higher incidence of multiple cancers. Given this high risk, initial treatment strategies should prioritize organ preservation and the ability to employ salvage treatments in the future. Surgery, particularly when RT is deferred, offers the advantage of preserving RT as a reserve option for second primary tumors or recurrence, which is highly relevant in a setting of high metachronous cancer incidence. In this context, it is also important to note that among patients with positive surgical margins in our study, the majority had horizontal (mucosal) margin involvement, and only one patient had both horizontal and vertical positivity, who was the only case to receive postoperative chemoradiotherapy. Local recurrence in patients with positive margins tended to occur relatively late, with a median time to recurrence of 1375 days (range: 168–4081). These findings support the clinical validity of close surveillance without immediate adjuvant therapy in selected early-stage cases, particularly in the absence of other high-risk pathological features such as deep margin involvement or lymphovascular invasion. This approach preserves RT for future salvage therapy and maintains options when multiple cancers appear in the head and neck region, consistent with the principles of organ preservation in head and neck oncology.

A limitation of this study is that it was not possible to show an association between p16 and survival, perhaps due to the small sample size, as this study included several older cases from the early 2000s that could not be investigated regarding human papillomavirus (HPV) status (Table 7). Also, although most patients in this study were p16-negative, global trends show that HPV-associated OPSCCs are steadily increasing, especially among the younger generation [39,40,41]. Since HPV-related OPSCCs are reported to have improved prognosis and survival in comparison to HPV-negative OPSCCs, regardless of the treatment modality [42], young patients with HPV-related OPSCCs require long-term perspectives in terms of treatment. RT tends to be selected for HPV-related OPSCCs. However, Ann et al. [10] reported that the incidence of multiple cancers in the head and neck region is still high, whether HPV status is negative or positive. Since younger patients can expect long-term survival, even if they have HPV-positive OPSCCs, surgery alone has the advantage of treatment for the metachronous multiple cancers. In this context, surgery may be especially appealing for younger, HPV-positive patients as it combines oncological effectiveness with functional preservation and defers radiation for future use if needed.

## 5. Conclusions

Surgical therapy is an effective treatment for T1/2N0 OPSCCs, considering the incidence of metachronous and multiple primary cancers.

## Figures and Tables

**Figure 1 cancers-17-01862-f001:**
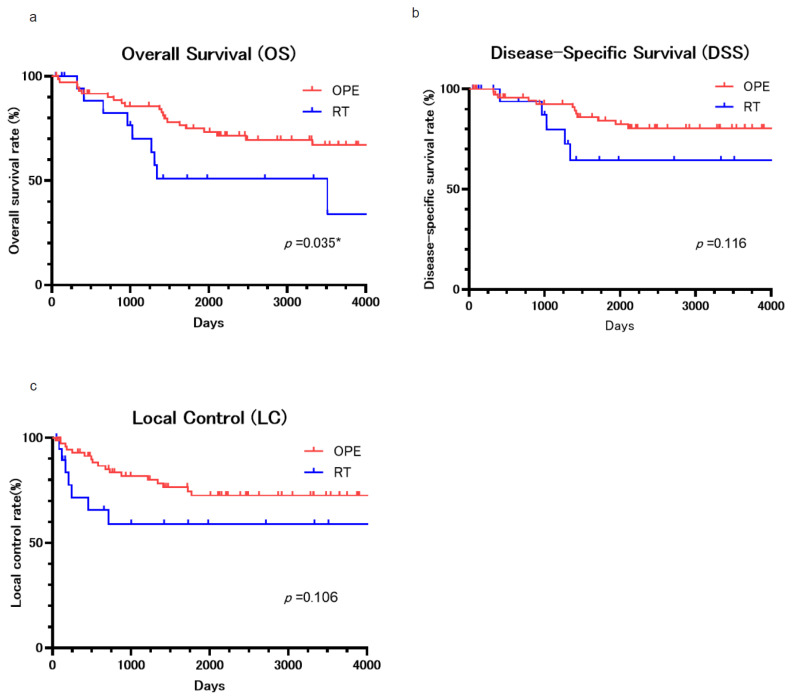
(**a**) The overall survival (OS) curve. All patients’ 5-year OS and 10-year OS rates were 70.2 and 61.2%. The 5-year OS was 74.9 and 51.0% for the surgery and RT groups, and the 10-year OS was 67.0 and 34.0% for the surgery and RT groups. (**b**) The disease-specific survival (DSS) curve. All patients’ 5-year DSS and 10-year DSS rates were 80.5 and 77.4%. The 5-year DSS was 86.0 and 64.5% for the surgery and RT groups, and the 10-year DSS was 80.5 and 64.5% for the surgery and RT groups. (**c**) The local control (LC) curve. The 5-year LC rates were 76.5 and 59.1% for the surgery and RT groups, respectively (Figure 1c). The 10-year LC rates were 72.6 and 59.1% for the surgery and RT groups, respectively. *: *p* < 0.05.

**Figure 2 cancers-17-01862-f002:**
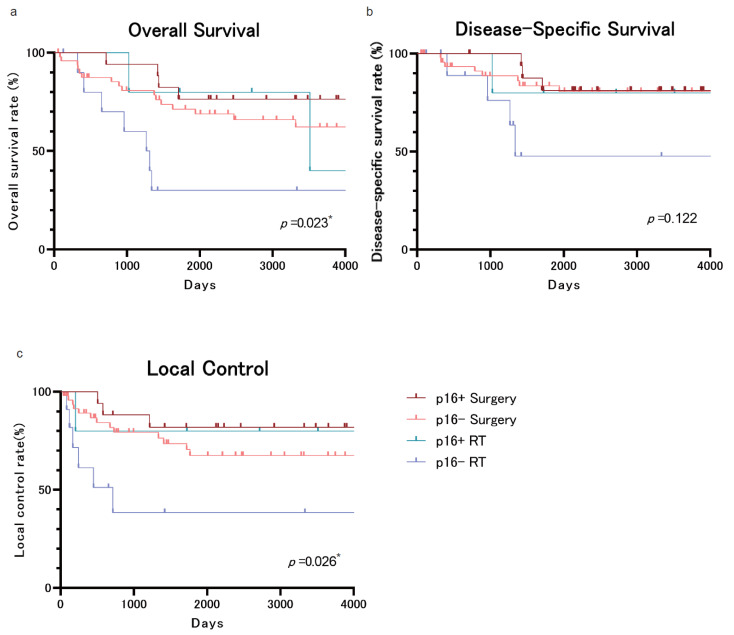
(**a**) The OS curve. The 5-year OS rates were 76.5%/71.4%/80.0%/30.0% for p16+ surgery/p16-surgery/p16+RT/p16-RT, respectively. (**b**) The DSS curve. The 5-year DSS rates were 81.3%/83.6%/80.0%/47.6% for p16+surgery/p16-surgery/p16+RT/p16-RT, respectively. (**c**) The LC curve. The 5-year LC rates were 81.9%/73.6%/80.0%/38.4% for p16+surgery/p16-surgery/p16+RT/p16-RT, respectively. *: *p* < 0.05.

**Table 1 cancers-17-01862-t001:** Details of the 94 patients who underwent surgery or radiotherapy.

Treatment				
	Surgery (*N* = 74)	RT (*N* = 20)	Total (*N* = 94)	*p*-Value
**Age** (median)	66.5 (35−82)	77.5 (49−92)	68 (35−92)	<0.001 ***
(range)				
**Gender** (%)				
male	61 (82.4)	19 (95.0)	80 (85.1)	0.288
female	13 (17.6)	1 (5.0)	14 (14.9)	
**Primary Tumor (T)** (%)				
T1	30 (40.5)	4 (20.0)	34 (36.2)	0.118
T2	44 (59.5)	16 (80.0)	60 (63.8)	
**8th-Edition Stage** (%)				
I	41 (55.4)	9 (45.0)	50 (53.2)	0.456
II	33 (44.6)	11 (55.0)	44 (46.8)	
**Subsite** (%)				
lateral	39 (52.7)	14 (70.0)	53 (56.4)	
anterior	10 (13.5)	2 (10.0)	12 (12.8)	0.348
upper	20 (27.0)	2 (10.0)	22 (23.4)	
posterior	5 (6.8)	2 (10.0)	7 (7.4)	
**p16 Status** (%)				
negative	50 (67.6)	11 (55.0)	61 (64.9)	
positive	17 (23.0)	5 (25.0)	22 (23.4)	0.38
unknown	7 (9.5)	4 (20.0)	11 (11.7)	
**Multiple Cancer** (%)				
none	32 (43.2)	8 (40.0)	40 (42.6)	1
multiple cancer	42 (56.8)	12 (60.0)	54 (57.4)	

RT: radiotherapy; ***: *p* < 0.001.

**Table 2 cancers-17-01862-t002:** Details of the 74 patients who underwent TOS or open resection.

Surgery				
	TOS (*N* = 57)	Open (*N* = 17)	Total (*N* = 74)	*p*-Value
**Age** (median)	67 (35−82)	66 (47−75)	66.5 (35−82)	0.689
(range)				
**Gender** (%)				
male	48 (84.2)	13 (76.5)	61 (82.4)	0.48
female	9 (15.8)	4 (23.5)	13 (17.6)	
**Primary Tumor (T)** (%)				
T1	27 (47.4)	3 (17.6)	30 (40.5)	0.047 *
T2	30 (52.6)	14 (82.4)	44 (59.5)	
**8th-edition Stage** (%)				
I	35 (61.4)	6 (35.3)	41 (55.4)	0.094
II	22 (38.6)	11 (64.7)	33 (44.6)	
**Subsite** (%)				
lateral	34 (59.6)	5 (29.4)	39 (52.7)	
anterior	1 (1.8)	9 (52.9)	10 (13.5)	<0.001 ***
upper	18 (31.6)	2 (11.8)	20 (27.0)	
posterior	4 (7.0)	1 (5.9)	5 (6.8)	
**p16 Status** (%)				
negative	37 (64.9)	13 (76.5)	50 (67.5)	
positive	14 (24.6)	3 (17.6)	17 (23.0)	0.755
unknown	6 (10.5)	1 (5.9)	7 (9.5)	
**Concurrent ND** (%)				
none	54 (94.7)	1 (5.9)	55 (74.3)	<0.001 ***
ND	3 (5.3)	16 (94.1)	19 (25.7)	
**Margin** (%)				
negative	31 (54.4)	11 (64.7)	42 (56.8)	
positive	26 (45.6)	6 (35.3)	32 (43.2)	0.580
**Positive Margin** (*N* = 32)				
horizontal	23 (88.5)	4 (66.7)	27 (84.5)	
vertical	2 (7.7)	0 (0.0)	2 (6.2)	0.112
both	1 (3.8)	2 (33.3)	3 (9.3)	

TOS: transoral surgery; Open: pharyngectomy via transcervical approach; ND: neck dissection; RT: radiotherapy. *p* values are shown (*: *p* < 0.05, ***: *p* < 0.001).

**Table 3 cancers-17-01862-t003:** The salvage treatments after the first recurrence and their results.

Recurrence (*N* = 40)	Initial Treatment			Total (*N* = 94)
	TOS (*N* = 57)	Open (*N* = 17)	RT (*N* = 20)	
**Primary lesion** (rT)	10 7 surgery (4 NER, 2 DOD, 1 DOC) 1 RT (DOD) 2 BSC (DOD)	7 5 surgery (4 NER, 1 DOD, 1 DOC) 1 RT (NER) 1 BSC (DOD)	7 6 surgery (5 DOD, 1 DOC) 1 BSC (DOD)	24 19 surgery 2 RT 4 BSC
**Regional lymph node** (rN)	14 13 surgery (7 NER, 4 DOD, 2 DOC) 1 RT (DOC)	1 1 surgery (NER)	0	15 14 surgery 1 RT
**Distant metastasis** (rM)	0	0	0	0
**All** (rTNM)	1 BSC (DOD)	0	0	1 BSC
**None**	32	9	13	54

RT: radiotherapy; TOS: transoral surgery; Open: pharyngectomy via transcervical approach; BSC: best supportive care. The numbers indicate the number of patients. NER: no evidence of recurrence; DOD: dead of disease; DOC: dead of other causes.

**Table 4 cancers-17-01862-t004:** The whole outcome following the initial treatment.

Initial Treatment					
	Outcome	TOS (*N* = 57)	Open (*N* = 17)	RT (*N* = 20)	Total (*N* = 94)
**NER**		38	15	11	64
**DOD**	Due to the primary lesion	5	1	5	11
	Due to the regional lymph node	2	0	0	2
	Due to the distant metastases	3	0	0	3
**DOC**	Malignant neoplasm	7	1	2	10
	Other than malignant neoplasm	2	0	2	4

RT: radiotherapy; TOS: transoral surgery; Open: pharyngectomy via transcervical approach; NER: no evidence of recurrence; DOD: dead of disease; DOC: dead of other causes. The numbers indicate the number of patients.

**Table 5 cancers-17-01862-t005:** Univariate analysis of the overall survival rate.

Factor	Hazard Ratio	95% Confidence Interval	*p*-Value
**Age**	1.041	0.996−1.087	0.073
**Gender**	1.044	0.399−2.732	0.931
**Primary Tumor (T)**	1.484	0.679−3.242	0.322
**8th-Edition Stage**	1.997	0.969−4.117	0.061
**p16 Status**	0.560	0.228−1.378	0.207
**Multiple Cancer**	2.62	1.124−6.107	0.025 *
**Treatment (Surgery or RT)**	2.271	1.034−4.987	0.041 *

*: *p* < 0.05.

**Table 6 cancers-17-01862-t006:** Multivariate analysis of the overall survival rate.

Factor	Hazard Ratio	95% Confidence Interval	*p*-Value
**Treatment (Surgery or RT)**	2.752	1.229−6.163	0.014 *
**8th-Edition Stage**	1.958	0.803−4.776	0.139
**p16 Status**	0.737	0.246−2.210	0.586

*: *p* < 0.05.

**Table 7 cancers-17-01862-t007:** Details of the sites of the multiple cancers.

Head and Neck Region		Other Regions	
Hypopharynx	17	Esophagus	24
Oral cavity	10	Stomach	12
Nasopharynx	3	Lung	7
Larynx	2	Colon	
Sinonasal	2	Pancreas	
		Prostate	
		Liver, Breast, Ovary for each	

## Data Availability

The datasets generated and/or analyzed during the current study are not publicly available due to privacy and ethical restrictions, but are available from the corresponding author on reasonable request and subject to institutional approval.

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
