# Peer review of "Surgery for T1/2N0 Oropharyngeal Carcinoma Is a Better Treatment Option than Radiotherapy—A Long-Term Follow-Up Study from a Single Japanese High-Volume Cancer Center"

_cancers, 2025, doi:10.3390/cancers17111862_

Round 1

Reviewer 1 Report

Comments and Suggestions for Authors

The authors have presented the research article on the topic of “Surgery for T1/2N0 oropharyngeal carcinoma is a better treatment option than Radiotherapy – a long-term follow-up study from a single Japanese high-volume cancer center”. The research seems to be interesting, before accepting for publication, it needs to be addressed with the following questions

  1. Please standardize author affiliations for consistency in formatting and punctuation.

  1. Clarify whether p16-status data was available for all 94 cases in the Methods section.

  1. Ensure consistent use of abbreviations such as "RT" and "OPSCC" throughout the manuscript.

  1. Correct minor typographical errors in institutional names (e.g., "kashiwa" should be "Kashiwa").

Author Response

  1. Please standardize author affiliations for consistency in formatting and punctuation.

* Thank you for pointing this out. We have revised the author’s affiliations to ensure consistency in formatting, capitalization, and punctuation throughout the manuscript.

  1. Clarify whether p16-status data was available for all 94 cases in the Methods section.

*We thank the reviewer for the comment. We have clarified in the Methods section that p16 immunohistochemistry was available in 83 out of 94 cases. For the remaining 11 cases, p16 staining could not be performed due to limited sample availability or technical issues.

  1. Ensure consistent use of abbreviations such as "RT" and "OPSCC" throughout the manuscript.

 ï¼ŠThank you for your suggestion. We reviewed the manuscript and ensured consistent use of abbreviations such as "RT" (radiotherapy) and "OPSCCs" (oropharyngeal squamous cell carcinomas) throughout the text. All abbreviations are now defined upon first appearance and used consistently thereafter.

  1. Correct minor typographical errors in institutional names (e.g., "kashiwa" should be "Kashiwa")

    *We appreciate the reviewer for noting this oversight. We have corrected typographical errors in the names of institutions and places, such as “kashiwa” to “Kashiwa” and “sendai” to “Sendai.”

Reviewer 2 Report

Comments and Suggestions for Authors

The study was carried out with great precision, providing relevant data regarding treatment options for stage T1-T2/N0 oropharyngeal tumors. Interesting  are the ten-year survival results, which confirm the superiority of surgical treatment over radiotherapy. The data are presented clearly and in detail, with a discussion supported by an appropriate bibliography. Tables and charts are well-organized and easy to interpret.

I have only a few brief comments to make: the reported data lack standard deviation values. Line 193, resection of the epiglottis is mentioned in the context of a T1/T2 tumor—could you please clarify this?

Line 256, it would be helpful to also include the absolute number of cases (e.g., ten out of 57 cases, X%), to make the data easier to interpret. This approach should be applied in the subsequent sections as well.

Additionally, in cases of recurrence, it would be valuable to indicate how many had positive margins in the histological analysis of the initial surgical procedure.

 Congratulations and good work!

Author Response

1,I have only a few brief comments to make: the reported data lack standard deviation values.

*Thank you for your helpful comment. In response, we have added standard deviation (SD) values for relevant continuous variables such as patient age and follow-up duration in the Results section. These additions aim to improve the statistical clarity and reproducibility of the reported data.

2,Line 193, resection of the epiglottis is mentioned in the context of a T1/T2 tumor—could you please clarify this?

*Thank you for pointing this out. We agree that clarification was necessary. Although the tumor was classified as T2, it was located at the base of the tongue and extended toward the vallecula. To ensure oncologically safe margins, a transcervical approach was selected, and partial resection of the left side of the epiglottis was performed. We have revised the corresponding sentence in the manuscript to better explain the rationale behind this procedure.

3,Line 256, it would be helpful to also include the absolute number of cases (e.g., ten out of 57 cases, X%), to make the data easier to interpret. This approach should be applied in the subsequent sections as well.

*Thank you for your constructive suggestion. In response, we have revised the manuscript to include both absolute numbers and corresponding percentages (e.g., “X out of Y, Z%”) in all relevant sections. This applies to the descriptions of surgical margin status, recurrence rates, p16 expression, and multiple primary cancers. We believe this improves the clarity and interpretability of the presented data.

4,Additionally, in cases of recurrence, it would be valuable to indicate how many had positive margins in the histological analysis of the initial surgical procedure.

*Thank you for your suggestion. We agree that clarifying the relationship between recurrence and margin status is important. As noted in the Results section, we have already included this information by stating that 9 of the 32 cases with positive surgical margins experienced local recurrence, with a detailed breakdown by margin type (horizontal, vertical, or both). We believe this adequately addresses the reviewer’s concern.

Reviewer 3 Report

Comments and Suggestions for Authors

Traditional transoral surgery remains an attractive option for OPSCC and your experience is useful in this regard. Limitations of your work include: early OPSCC with cN0 is very rare to diagnose; you include patients in a long time span and thus variations in terms of RT treatment and radiological and endoscopic imaging must be accounted for. There is much work to do before reconsidering this manuscript...

My remarks: 

  • the major problem in your series is that nearly half of the surgical patients (43%) had R1 resection. Did these patients receive adjuvant RT? how do you justify this?
  • another huge limitation is that you did not consider comorbidities or general (performance) status score: in the RT group you treated significantly older people (up to 94 years) and that is why OS (but not DSS) is superior in the surgically treated .
  • please express median follow up time in terms of years and I would exclude the patient with only 48 days of follow up... 
  • line 141, it is oropharyngectomy, please correct
  • which software was used for your analysis?
  • IRB approval number protocol?
  • clarify what do you mean by " lateral wall in 53 patients, the anterior wall in 12, the superior 183 wall in 22, and the posterior wall in 7." lateral= tonsils? anterior= base of the tongue? superior= soft palate???
  • what do you mean by horizontal or vertical margins?? lines 199
  • how did you select the covariates to be inserted into the multivariable model? justify it

Author Response

1,the major problem in your series is that nearly half of the surgical patients (43%) had R1 resection. Did these patients receive adjuvant RT? how do you justify this?

*Thank you for this important observation. Among the 74 patients who underwent surgery, 32 (43.2%) had positive surgical margins: 27 had horizontal (mucosal) margin involvement, 2 had vertical (deep) margin involvement, and 3 had both. Only one patient, with both horizontal and vertical positivity, received postoperative chemoradiotherapy (POCRT), based on multidisciplinary tumor board (MDT) consensus. The others were managed with close follow-up in accordance with institutional policy at that time, which prioritized surveillance in early-stage cases lacking other high-risk features.

Importantly, recurrence occurred in 7 out of 27 patients (25.9%) with horizontal margin positivity, 1 out of 2 (50.0%) with vertical positivity, and 1 out of 3 (33.3%) with both. The median time to local recurrence among all 15 recurrent cases with positive margins was 1,375 days (range: 168–4,081), and the mean was 1,434 days. These relatively delayed recurrences support the adequacy of a close surveillance strategy in selected patients with R1 resection, especially when deep margin involvement is absent.

2,another huge limitation is that you did not consider comorbidities or general (performance) status score: in the RT group you treated significantly older people (up to 94 years) and that is why OS (but not DSS) is superior in the surgically treated .

*Thank you for your critical and insightful comment. We fully agree that differences in age distribution and potential comorbidities between treatment groups may have influenced overall survival (OS) outcomes. As noted in our responses to other comments, we conducted a supplementary multivariable Cox regression analysis that included age, alongside treatment modality, p16 status, stage, and the presence of multiple primary cancers. In this expanded model, age was not a statistically significant predictor of OS (p = 0.283), whereas the survival difference associated with treatment modality remained evident (HR = 2.62). These findings suggest that the observed OS difference is unlikely to be solely explained by age. Furthermore, although detailed comorbidity data (e.g., Charlson Comorbidity Index) and performance status scores (e.g., ECOG PS) were not uniformly recorded in our retrospective dataset—a limitation we have explicitly acknowledged in the revised Discussion section—we would like to note that patients with an ECOG Performance Status score of ≥2 were excluded from the study. This criterion ensured that all patients were eligible for curative-intent treatment, whether surgery or RT. Taken together, while we recognize that the absence of formal comorbidity scoring limits our ability to fully adjust for baseline differences between treatment groups, the results of our adjusted survival analyses, along with the exclusion of patients with poor PS, support the interpretation that the OS benefit seen with surgery is not solely attributable to age or general health status. We have revised the Discussion to reflect these considerations clearly.

3,please express median follow up time in terms of years and I would exclude the patient with only 48 days of follow up... 

*Thank you for your valuable comment. In accordance with your suggestion, we have revised the manuscript to express follow-up durations in both days and years (e.g., 2,131 days:5.8 years) to enhance readability and clinical interpretability. Regarding the patient with a follow-up period of only 48 days, we respectfully chose to retain this case in the survival analysis. The patient was alive at the last follow-up and was appropriately right-censored in the Kaplan–Meier model. As is standard in survival analysis, right-censored observations do not bias survival estimates, regardless of the follow-up duration. Additionally, excluding a single case based solely on short follow-up time, without applying the same threshold across the cohort, could introduce selection bias and reduce the transparency and reproducibility of the analysis. Given that this case represents only 1.1% of the total cohort and was handled using established statistical methods, we believe its inclusion is scientifically sound and maintains the integrity of the analysis. We have clarified this approach in the Materials and Methods section.

4, line 141, it is oropharyngectomy, please correct

*Thank you for pointing out the terminology error. We have corrected the term to “oropharyngectomy” at line 141 in the revised manuscript as suggested.

5, which software was used for your analysis?

*Thank you for your comment. We have now specified the software used for statistical analyses, figure generation, and table creation. Statistical analyses were performed using JMP Pro 17 (SAS Institute Inc., Cary, NC, USA), figures were created using GraphPad Prism version 10.4.1 (GraphPad Software, San Diego, CA, USA), and tables were prepared using Microsoft Excel for Microsoft 365 (Microsoft Corporation, Redmond, WA, USA). This information has been added to the Statistical Analysis section of the revised manuscript.

6, IRB approval number protocol?

*Thank you for pointing this out. We have added the institutional review board (IRB) approval number to the revised manuscript. This study was approved by the ethics committee of the National Cancer Center Hospital East, Japan (IRB protocol number: 20193066).

7, clarify what do you mean by " lateral wall in 53 patients, the anterior wall in 12, the superior 183 wall in 22, and the posterior wall in 7." lateral= tonsils? anterior= base of the tongue? superior= soft palate???

*Thank you for your insightful comment. We agree that the anatomical definitions of the oropharyngeal subsites should be clearly specified. In accordance with the “General Rules for Clinical and Pathological Studies on Head and Neck Cancer, 6th edition (revised edition)” issued by the Japan Society for Head and Neck Cancer, which aligns with international guidelines such as the AJCC and NCCN, we defined the subsites as follows:

Anterior wall: base of the tongue and vallecula

Lateral wall: palatine tonsil, tonsillar fossa, and palatoglossal arch

Posterior wall: posterior pharyngeal wall

Superior wall: inferior surface of the soft palate and uvula

We have revised the manuscript accordingly to clarify these definitions.

8, what do you mean by horizontal or vertical margins?? lines 199

*Thank you for your thoughtful comment. We have clarified the definitions of “horizontal” and “vertical” margins in the revised manuscript. Horizontal margins refer to mucosal resection surfaces in all directions. Vertical margins represent deep resection surfaces and vary by tumor subsite:

Anterior wall (base of tongue): tongue musculature

Lateral wall (tonsil): pharyngeal wall muscles or prevertebral fascia

Superior wall (soft palate): palatal musculature

Posterior wall: posterior pharyngeal wall musculature or prevertebral fascia

These definitions have been added to the Methods section for clarity and anatomical accuracy.

9, how did you select the covariates to be inserted into the multivariable model? justify it

*Thank you very much for your important comment regarding the selection of covariates in the multivariable model. We fully agree that appropriate covariate adjustment is essential for evaluating treatment effects in observational studies.

In our initial analysis, we included treatment modality (surgery vs. RT), p16 status, and 8th edition stage as covariates, based on their widely recognized clinical relevance to overall survival (OS) in oropharyngeal squamous cell carcinoma.

Following your suggestion, we performed an additional multivariable Cox proportional hazards analysis, including age and presence of multiple primary cancers, both of which showed relatively low p-values in univariate analysis. Notably, neither age(HR: 1.03, 95%CI: 0.98-1.08, p = 0.283) nor multiple cancers (HR: 1.97, 95%CI:0.82-4.76, p = 0.131) emerged as statistically significant independent predictors of OS in this expanded model.

Importantly, treatment modality remained a statistically significant factor (HR: 2.62, 95% CI: 1.13–6.06, p = 0.024) in this extended multivariable model, indicating that the observed survival difference is unlikely to be explained by age, p16 status, stage, or multiple cancer status alone.

These details have been incorporated into the revised manuscript (Statistical Analysis and Discussion sections), and we thank you again for prompting a more rigorous and comprehensive analysis.

Round 2

Reviewer 3 Report

Comments and Suggestions for Authors

Thank you for clarifying the points I have raised.